# Towards Pareto-Optimality for Test-Time Adaptation

## Abstract

Test-Time Adaptation (TTA) has been effective for mitigating the distribution shifts of test datasets by adapting a pre-trained model. The existing TTA approaches update the model parameters online toward the gradient descent direction by averaging individual objectives in the current batch. The averaged gradient can be biased by only a few instances in the batch, leading to conflict among individual objectives when updating the model. To prevent a negative effect from the gradient conflict among test instances, a model could have been updated by the gradient that is agreeable by all objectives in the batch. Therefore, we propose a new approach to update the model parameters toward Pareto-Optimality across all individual objectives in TTA. Particularly, this paper suggests an extended version of the Pareto optimization to anticipate unexpected distribution shifts during testing time. This extension is enabled by merging the sharpness-aware minimization into the Pareto optimization. We demonstrate the effectiveness of the proposed approaches through experiments on three benchmark datasets: CIFAR10-to-CIFAR10C, CIFAR100-to-CIFAR100C, and ImageNet-to-ImageNetC.

## 1 Introduction

Deep neural networks (DNNs) have achieved successful performance in many applications (He et al., 2016; Wang et al., 2018). However, the trained DNNs suffer from the performance degradation when there are distribution shifts between training data and test data (Ganin & Lempitsky, 2015; Zhou et al., 2022a). In real-world applications, these distribution shifts often appear as environmental changes, including natural phenomena such as weather conditions (e.g., snow, brightness, or fog), sensor-related noise (e.g., Gaussian noise, impulse noise, or glass blur), etc (Hendrycks & Dietterich, 2019). These factors are usually unseen when training the models, so the pretrained model often leads to unexpected decisions under this distribution shift. Therefore, given the knowledge from a trained model, it is necessary to adapt the model effectively to perform well on such unseen test data. In addition to distribution shifts, due to privacy concerns or memory issues, the training data is generally unavailable during the inference phase of testing (Liang et al., 2020). Furthermore, the model adaptation needs to be performed in an online manner, i.e, streamed test data, because testing environment often faces a series of queries for model outputs (Wang et al., 2020).

Given the necessity of model adaptation, Test-Time Adaptation (TTA) is proposed recently to mitigate the distribution shifts under these constrained scenario (Sakaridis et al., 2021; Wang et al., 2022), and TTA has shown remarkable performance improvement. The goal of TTA is to adapt the source-trained model to perform well on the unseen test data during the testing phase on the fly. Specifically, current TTA approaches design objectives, such as entropy loss on the test data (Wang et al., 2020); and these approaches update the model parameters by aggregating individual objectives in the batch. This batch-wise model updating is favored for its efficiency and practicality. However, it is well-known that the averaged gradients greedily minimize the batch-wise learning loss (Parascandolo et al., 2020). The averaged gradient direction can be biased only by a subset of instances in the batch, potentially causing conflict to individual objectives. Therefore, TTA needs to find the updating direction to ensure effective adaptation across all instances in the batch without conflicts, rather than being influenced mainly by a subset of instances.

This paper provides the first empirical result of an average gradient from a batch causing conflicts among individual objectives. We also show the degradation of adaptation performance because

of this conflict. To address this problem, we propose a new gradient-based update approach for TTA, that aims to improve the performance of all individual objectives in the batch. We leverage Pareto-Optimality across the individual objectives in the batch, guiding the model towards a state where none of the individual objectives can be improved further. The Pareto-oriented gradient-based update leads to reduce the individual conflicts, thereby preventing model collapse due to error accumulation during online updating.

Whereas Pareto optimization reduces these conflicts from individual gradients, we strengthen this optimization also by seeking the flat-minima from the Pareto-optimization gradients. The online updating of TTA models can exhibit instability in extreme scenarios, including mixed distribution shifts, label distribution shifts, or small batch sizes (Niu et al., 2023). Inspired by Sharpness-Aware Minimization (SAM) (Foret et al., 2020), TTA approach aimed at seeking flat minima have been proposed in order to address these extreme cases, called SAR (Niu et al., 2023). Fundamentally, SAR identifies the direction of maximal perturbation from the averaged objective, i.e., by averaging individual entropy losses. However, the maximally perturbed direction from the averaged objective is still susceptible to conflicts among individual instances, which hinder to find flatness of the individual objectives appropriately. Therefore, when we consider flatness of the individual objectives in the batch, we need to find the perturbation direction without the conflicts among instances. To address this limitation, we propose a sharpness-aware Pareto-oriented gradient optimization.

We summarize our contributions as follows. We claim that the existing approaches aggregate the individual objectives in the batch without considering conflicts, resulting in the biased direction. Therefore, we introduce Pareto-Optimality to improve all individual objectives from updating the model. Furthermore, combined with SAM approach, we propose a new approach to find the perturbed direction in order to maximize all individual objectives, not an aggregated objective from individuals. We demonstrate the effectiveness of the proposed approaches through experiments on benchmark datasets for classification tasks.

## 2 PRELIMINARY

### 2.1 ENTROPY-BASED TEST-TIME ADAPTATION

This section starts by introducing notations. Let $D_S = \{(x_s^i, y_s^i)\}_{i=1}^{n_s}$ be the dataset from source domain; and let $D_T = \{(x_t^i, y_t^i)\}_{i=1}^{n_t}$ be the dataset from the target domain where $\{y_t^i\}_{i=1}^{n_t}$ is not available. $D_S$ and $D_T$ are sampled from two different distributions of a source domain, $p_s(x, y)$, and a target domain, $p_t(x, y)$, respectively. $f_\theta(x)$ is a model with parameters $\theta$, trained by a source dataset, $D_S$. TTA assumes that we only have access on a pre-trained model $f_\theta$ without the source dataset $D_S$ used for the training. Therefore, the goal of TTA is adapting the source-trained model to the target domain during the testing phase on the fly. Since label information for the target domain is not available, one possible objective of TTA is to minimize the entropy of its predictions from the source-trained model. We summarize such entropy-based approach, TENT (Wang et al., 2020).

We denote an instance $x_t^i$ from the target dataset as $x_i$ for simplicity, due to the absence of source datasets. The objective of entropy loss, $\mathcal{L}(\theta)$, with the batch size of $B$ is defined as,

$$\min_\theta \mathcal{L}(\theta) = \min_\theta \sum_{i=1}^{B} \frac{1}{B} \ell(x_i; \theta), \tag{1}$$

where $\ell(x_i; \theta)$ represents an individual entropy loss for an instance $x_i$. TENT (Wang et al., 2020) then optimizes the model parameters to minimize $\mathcal{L}(\theta)$ that is the averaged entropy loss from individual test instances, so the minimized entropy will finetune the decision boundary to be more clear. Gradient-based optimization (Amari, 1993) updates the model parameter $\theta$ in the descent direction of $d$ as follows:

$$\theta' \leftarrow \theta - \eta d \quad \text{where} \quad d = \sum_{i=1}^{B} \frac{1}{B} \nabla_\theta \ell(x_i; \theta). \tag{2}$$

Here, $\nabla_\theta \ell(x_i; \theta)$ is the gradient of an individual entropy loss with respect to the model parameter $\theta$.

### 2.2 SHARPNESS-AWARE MINIMIZATION

Understanding the loss landscape plays an important role in training a model for the perspective of generalization ability (Jiang et al., 2019). Especially, optimizing model parameters toward flat min-

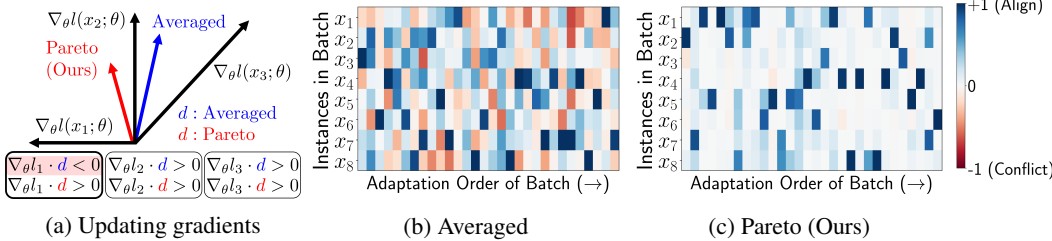

(a) Updating gradients      (b) Averaged      (c) Pareto (Ours)

Figure 1: (a) Illustration of relationship between the individual gradients and updating gradients using Averaged and Pareto (Ours) approaches, respectively. (b)-(c) Visualization of gradient conflicts analyzed through cosine similarity between the updating gradients and individual gradients in the batch. (red represents the conflict with individual objective due to the updating direction, while blue denotes alignment.)

ima is known to improve model robustness against distribution shifts (Jiang et al., 2019). Recently, sharpness-aware minimization (SAM) has received great attention because of its effectiveness and scalability Foret et al. (2020). Inspired by the relationship between the population risk and the worst-case training loss in the neighborhood, SAM proposes the following optimization problem:

$$\min_{\theta} \mathcal{L}_{SAM}(\theta) + \gamma\|\theta\|_2^2 \quad \text{where} \quad \mathcal{L}_{SAM}(\theta) = \max_{\|\epsilon\|_p \leq \rho} \mathcal{L}(\theta + \epsilon), \tag{3}$$

where $\rho > 0$ is the maximum magnitude of the perturbation, $\gamma > 0$ is an adjusting hyperparameter of the weight decay term, and $p \in [1, \infty]$ is the degree of the norm. To efficiently solve the inner optimization, SAM utilizes the first-order Taylor approximation of $\mathcal{L}(\theta + \epsilon)$:

$$\hat{\epsilon}(\theta) = \arg\max_{\|\epsilon\|_p \leq \rho} \mathcal{L}(\theta + \epsilon) \approx \arg\max_{\|\epsilon\|_p \leq \rho} \left\{ \mathcal{L}(\theta) + \epsilon^T \nabla_\theta \mathcal{L}(\theta) \right\} = \arg\max_{\|\epsilon\|_p \leq \rho} \epsilon^T \nabla_\theta \mathcal{L}(\theta) \tag{4}$$

By doing this approximation, we can get the closed form of $\hat{\epsilon}$ (Equation 2 in Foret et al. (2020)). Given the maximally perturbed direction $\hat{\epsilon}$, the original objective of SAM can be written as follows:

$$\min_{\theta} \mathcal{L}(\theta + \hat{\epsilon}) + \gamma\|\theta\|_2^2. \tag{5}$$

### 2.3 PARETO-OPTIMALITY IN MULTI-OBJECTIVE OPTIMIZATION

Multi-objective optimization is an optimization on multiple objective functions that possibly have conflicts between two or more objectives. Since there will be trade-offs in different efficiency between conflicting objectives, we can not greedily optimize each of objective functions, individually. Therefore, the goal of multi-objective optimization is finding solutions to improve a set of objectives without reducing the other objectives, which is called Pareto-Optimality. Pareto-Optimality is widely used in many machine learning areas such as Multi-task learning (Sener & Koltun, 2018; Lin et al., 2019; Liu et al., 2021), Reinforcement learning (Pirotta & Restelli, 2016; Yang et al., 2019), and Bayesian optimization (Belakaria et al., 2020; Konakovic Lukovic et al., 2020).

If we narrow the multi-objective optimization into adapting three testing instances, Figure 1a shows the illustrated concept of Pareto-Optimality. Let's assume that we have three black directions of gradient signals from individual test data instances. Their average direction becomes the blue arrow, which have the negative cosine similarity with one of the gradient: $\nabla_\theta \ell_1 \cdot d < 0$. This negative cosine similarity indicates its test instance to be reduced in its testing performance. Meanwhile, our suggestion, as red arrow, has positive cosine similarity with gradients from all test instances.

## 3 METHODOLOGY

### 3.1 MOTIVATION

The existing approaches for TTA update the model parameters toward the gradient descent direction by averaging individual gradients in the current batch (Niu et al., 2022; 2023). However, the averaged gradient can be biased toward only few instances in the batch, which makes conflict among individual gradients (Parascandolo et al., 2020). If there is a conflict between the updating gradient

direction $d$ and individual gradients of $\nabla_\theta \ell(x_i; \theta)$, this conflict causes a negative effect to adapt the test instances causing the conflict, as illustrated at Figure 1a. Therefore, updating the model should have a direction to adapt well on all instances in the batch. Figure 1 represents the cosine similarity between the updating direction $d$ and the individual gradients in the batch. Figure 1b shows that the averaged gradient leads to conflicts (red color) to many individual gradients. This conflict increases the entropy loss of Eq. 1 due to the negative cosine similarity. This means that the direction $d$ is not able to adapt the instances with conflict in the batch. Given this motivation, we need to find the gradient direction to reduce conflict to individual gradients in order to adapt well. Therefore, we propose a new way of gradient-based optimization for TTA, which is explained in the next section. We note that the proposed approach is applicable in existing TTA algorithms as an independent and orthogonal contribution.

## 3.2 Pareto Optimality for Test-Time Adaptation

We proposed to find the updating direction for minimizing all individual entropy losses in the update. TTA has a streamlined test batches whose adaptation will change the long-trained original model parameters. Therefore, this paper suggests to adapt the model at the test-time by the gradient-based optimization without conflicts among individual test losses in the batch.

The Pareto optimization is one of such conflict-free optimization on model parameters. As we do not assume to store any data instances from source or previous test datasets, we consider the Pareto optimization on the individual losses from a working test batch. Therefore, we define the objective of test-time adaptation from the aggregated loss, defined as Eq. 1, to vector-valued loss for the Pareto optimization in the below.

$$\min_\theta \mathbb{L}(\theta) = \min_\theta [\ell(x_1; \theta), \ell(x_2; \theta), ..., \ell(x_B; \theta)]^\top \tag{6}$$

**Definition 3.1 (Pareto-Optimality)** *For any two parameters $\theta$ and $\theta'$, we say that $\theta$ dominates $\theta'$, denoted as $\theta \prec \theta'$, if $\ell(x_i; \theta) \leq \ell(x_i; \theta')$ for all $i$ and $\mathbb{L}(\theta) \neq \mathbb{L}(\theta')$. A parameter $\theta^*$ is called Pareto-Optimal if there exists no solution $\theta$ that dominates $\theta^*$.*

Based on the Definition 3.1, we need to find the updating direction towards Pareto-Optimality to minimize all individual entropy losses. There is a necessary condition for Pareto-Optimality, which is called Pareto-criticality (Zhou et al., 2022b).

**Definition 3.2 (Pareto-Criticality)** *A parameter $\theta^*$ is called Pareto-Critical if there is no common gradient descent direction $d$ such that $\nabla_\theta \ell(x_i; \theta^*)^T d < 0$ for all $i$.*

From the definition, if a parameter is not positioned at the Pareto-critical point, there exists a descent direction that improves the objective of all instances, entropy loss in this paper. Therefore, we design a gradient-based optimization for finding Pareto-critical point. Following MGDA (Sener & Koltun, 2018), we find a Pareto-critical point by solving the subproblem:

$$(\hat{w}_1, ..., \hat{w}_B) = \underset{w_1, w_2, ..., w_B}{\arg\min} \left\{ \left\| \sum\nolimits_{i=1}^{B} w_i \nabla_\theta \ell(x_i; \theta) \right\|_2^2 \ \Big| \ \sum\nolimits_{i=1}^{B} w_i = 1, w_i \geq 0 \ \forall i \right\} \tag{7}$$

MGDA (Sener & Koltun, 2018) showed that, the optimal solution of this subproblem is 0 when $\theta$ is the Pareto-critical point; otherwise, the weighted sum of the gradients by optimal weights of this subproblem, $\sum_{i=1}^{B} \hat{w}_i \nabla_\theta \ell(x_i; \theta)$, is a common descent directions of tasks. This indicates that the weighted sum of gradients does not conflict with the gradients of individual tasks. Therefore, we apply a gradient descent step that utilizes this weighted sum of gradients toward the Pareto-critical point by optimizing the subproblem of Eq. 7:

$$\theta' \leftarrow \theta - \eta d \quad \text{where} \quad d = \sum\nolimits_{i=1}^{B} \hat{w}_i \nabla_\theta \ell(x_i; \theta). \tag{8}$$

Compared to Eq. 2, the proposed gradient direction of $d$ finds the adaptive individual weights $\hat{w}_i$ for reducing the conflicts between individual losses (see Figure 1c).

## 3.3 Problem of Sharpness-Aware Minimization for Test-Time Adaptation

Our proposed direction is effective in mitigating the conflicts when updating the model. Nevertheless, when dealing with real-world problems in TTA, the online updating of TTA models can become

unstable due to wild distribution shifts. Consequently, SAM-based TTA approach, called SAR, has been introduced to address these distribution shifts by aiming to seek flat minima in TTA objectives (or losses) (Niu et al., 2023). Fundamentally, SAR identifies the direction of maximal perturbation in the loss, averaged across instances, and then optimize the model parameters to minimize the loss resulting from these perturbed parameters.

In spite of the recent study of flat minima by SAR, this method is still susceptible to the concerns raised in Section 3.2: SAR do not account for the conflicts among individual instances. This becomes particularly problematic in the scenario of TTA because TTA has a few instances to learn from, so the conflicts among instances need to be resolved. In detail, identifying the perturbation direction based on the averaged loss does not maximize the loss for all individual instances. In other words, the perturbation vector by SAR may conflict to the gradient from individual losses, which calls for the expansion of Pareto-gradient optimization to the study of optimal perturbation. In light of this limitation, we propose an alternative approach known as sharpness-aware Pareto-oriented gradient optimization.

## 3.4 Sharpness-Aware Pareto Optimality for Test-Time Adaptation

This paper formulates the sharpness-aware Pareto optimization, and this is the first proposal of merging two concepts of flat-minima and Pareto optimality, to our knowledge. The suggested objective is:

$$\min_{\theta} \max_{\|\epsilon\| \leq \rho} \mathbb{L}(\theta + \epsilon) = \min_{\theta} \max_{\|\epsilon\| \leq \rho} [\ell(x_1; \theta + \epsilon), \ell(x_2; \theta + \epsilon), ..., \ell(x_B; \theta + \epsilon)]^{\top}. \tag{9}$$

The maximally perturbed direction $\epsilon$, calculated using the averaged loss, can be biased towards a few instances, as discussed in Section 3.2. Consequently, there is no guarantee that this direction maximizes all individual losses on the perturbed parameters. Thus, it motivates us to seek a perturbation direction that maximizes all individual losses. We propose to formulate a subproblem to find $\epsilon$ toward Pareto-Optimality for improving (maximizing) individual losses as below,

$$(\hat{\alpha}_1, ..., \hat{\alpha}_B) = \underset{\alpha_1, \alpha_2, .., \alpha_B}{\arg\min} \left\{ \left\| \sum_{i=1}^{B} \alpha_i \nabla_{\epsilon} \ell(x_i; \theta + \epsilon) \right\|_2^2 \; \Big| \; \sum_{i=1}^{B} \alpha_i = 1, \alpha_i \geq 0 \; \forall i \right\}. \tag{10}$$

Eq. 10 follows the same way with Eq. 7, but the individual objective is different, as $\epsilon$, $\nabla_{\epsilon}\ell(x_i; \theta + \epsilon)$. First, we solve the subproblem of Eq. 10 to find $\hat{\alpha}_i$. Then, we determine the direction $\hat{\epsilon}$ by applying the constraints for perturbation boundness in $\rho$, as below,

$$\hat{\epsilon} = \rho \frac{\Sigma_{i=1}^{B} \hat{\alpha}_i \nabla_{\epsilon} \ell(x_i; \theta + \epsilon)}{\|\Sigma_{i=1}^{B} \hat{\alpha}_i \nabla_{\epsilon} \ell(x_i; \theta + \epsilon)\|}. \tag{11}$$

We call a Pareto-oriented maximally perturbation direction $\hat{\epsilon}$, which is not explored even in fields of SAM researches. Consequently, we re-formulate a vector-valued perturbation loss to incorporate individual perturbed loss, as below,

$$\min_{\theta} \mathbb{L}(\theta + \hat{\epsilon}) = \min_{\theta} [\ell(x_1; \theta + \hat{\epsilon}), \ell(x_2; \theta + \hat{\epsilon}), ..., \ell(x_B; \theta + \hat{\epsilon})]^{\top}. \tag{12}$$

As we propose the Pareto-oriented gradient updates to optimize the model without conflicts, this optimization minimizes the model's perturbed loss in all individual instances without conflict. Therefore, we find the update direction toward Pareto-critical point over the individual perturbed loss,

$$\theta' \leftarrow \theta - \eta d \quad \text{where} \quad d = \sum_{i=1}^{B} \hat{w}_i \nabla_{\theta} \ell(x_i; \theta + \hat{\epsilon}), \tag{13}$$

where we calculate $\hat{w}_i$ by solving a subproblem as we design in Eq. 7 by replacing $\nabla_{\theta}\ell(x_i; \theta)$ to $\nabla_{\theta}\ell(x_i; \theta + \hat{\epsilon})$. It should be noted that the objective of the perturbed loss from SAM can be considered as the summation of sharpness and original loss, i.e., $\max_{\|\epsilon\| \leq \rho} \ell_i(\theta + \epsilon) = \{\max_{\|\epsilon\| \leq \rho} \ell_i(\theta + \epsilon) - \ell_i(\theta)\} + \ell_i(\theta)$. Therefore, we optimize the model to minimize both sharpness and task loss of individual instances without conflicts.

## 3.5 Training Algorithms

In order to find the proposed gradient update direction, we need to solve a sub-problem of Eq. 7 and Eq. 10 towards the Pareto-critical point. Following the approach in MGDA (Sener & Koltun, 2018),

Table 1: Accuracy (%) on ImageNet-C (severity level 5) under **standard online adaptation** with ResNet50-GN. The improvement by the proposed approaches are marked as bold in each corruption. The underline represents the best results on each corruption case. (Avg.: Average performance over all corruptions.)

| Method | Noise | | | Blur | | | | Weather | | | | Digital | | | | Avg. |
|---|---|---|---|---|---|---|---|---|---|---|---|---|---|---|---|---|
| | Gauss. | Shot | Impul. | Defoc. | Glass | Mot | Zoom | Snow | Frost | Fog | Brit. | Contr. | Elastic | Pixel | JPEG | |
| Source | 22.1 | 23.0 | 22.0 | 19.8 | 11.4 | 21.5 | 25.0 | 40.3 | 47.0 | 34.0 | 68.8 | 36.2 | 18.5 | 29.2 | 52.6 | 31.4 |
| TENT | 23.8 | 28.2 | 25.1 | 15.1 | 8.0 | 21.8 | 22.7 | 26.6 | 33.1 | 3.6 | 69.8 | 42.3 | 10.8 | 48.5 | 54.5 | 28.9 |
| **w/ Pareto** | 21.6 | 24.7 | 23.8 | 16.4 | 10.6 | 22.2 | 24.8 | 36.5 | **39.6** | 14.3 | 69.3 | 41.6 | 15.6 | 45.4 | 54.3 | 30.7±0.1 |
| **w/ ParetoSAM** | **25.8** | **31.5** | **27.9** | 14.9 | **12.3** | **25.3** | 25.4 | **37.6** | 39.4 | **25.1** | 69.3 | **43.9** | **16.0** | 45.8 | **54.6** | **33.0**±0.2 |
| SAR | 32.1 | 34.3 | 33.4 | 18.6 | 19.3 | 30.4 | 30.8 | 42.1 | 43.2 | 46.1 | 70.2 | 43.9 | 15.7 | 49.1 | 55.4 | 37.6 |
| **w/ ParetoSAM** | **41.2** | **43.5** | **42.5** | 17.9 | **20.6** | **39.8** | **41.2** | **49.2** | **47.7** | 34.7 | 69.1 | **48.7** | **31.0** | 46.0 | 54.0 | **41.8**±0.4 |

Table 2: Accuracy (%) on ImageNet-C (severity level 5) under **online imbalanced label shifts** with ResNet50-GN. The improvement by the proposed approaches are marked as bold in each corruption. The underline represents the best results on each corruption case. (Avg.: Average performance over all corruptions.)

| Method | Noise | | | Blur | | | | Weather | | | | Digital | | | | Avg. |
|---|---|---|---|---|---|---|---|---|---|---|---|---|---|---|---|---|
| | Gauss. | Shot | Impul. | Defoc. | Glass | Mot | Zoom | Snow | Frost | Fog | Brit. | Contr. | Elastic | Pixel | JPEG | |
| Source | 17.9 | 19.9 | 17.9 | 19.7 | 11.3 | 21.3 | 24.9 | 40.4 | 47.4 | 33.6 | 69.2 | 36.3 | 18.7 | 28.4 | 52.2 | 30.6 |
| MEMO | 18.4 | 20.6 | 18.4 | 17.1 | 12.7 | 21.8 | 26.9 | 40.7 | 46.9 | 34.8 | 69.6 | 36.4 | 19.2 | 32.2 | 53.4 | 31.3 |
| TENT | 2.6 | 3.3 | 2.7 | 13.9 | 7.9 | 19.5 | 17.0 | 16.5 | 21.9 | 1.8 | 70.5 | 42.2 | 6.6 | 49.4 | 53.7 | 22 |
| **w/ Pareto** | **18.9** | **19.8** | **20.0** | 14.4 | **8.5** | 19.3 | 22.4 | 34.6 | **36.3** | 13.5 | 69.3 | 41.3 | **15.2** | 46.4 | **54.2** | **28.9**±0.2 |
| **w/ ParetoSAM** | 18.7 | **19.8** | 18.9 | **14.5** | **8.5** | 18.8 | **22.5** | **34.8** | 35.6 | **14.4** | 69.3 | 41.1 | 14.9 | 45.9 | 54.1 | 28.8±0.1 |
| EATA | 27.0 | 28.3 | 28.1 | 14.9 | 17.1 | 24.4 | 25.3 | 32.2 | 32.0 | 39.8 | 66.7 | 33.6 | 24.5 | 41.9 | 38.4 | 31.6 |
| SAR | 33.1 | 36.5 | 35.5 | 19.2 | 19.5 | 33.3 | 27.7 | 23.9 | 45.3 | 50.1 | 71.9 | 46.7 | 7.1 | 52.1 | 56.3 | 37.2 |
| **w/ ParetoSAM** | **41.7** | **43.0** | **43.2** | 15.4 | 13.7 | **40.4** | **30.7** | **36.2** | **46.3** | 44.9 | 69.7 | **51.7** | 12.5 | **53.6** | **56.6** | **40.0**±0.4 |

we employ Frank-Wolfe algorithm to solve the constrained optimization problem (Jaggi, 2013). Consequently, the solver's computations cause additional computation time, which is proportional to the number of output dimensions, corresponding to the batch size $B$ in our case. There are some techniques to reduce the computations in multi-objective optimization fields (Liu et al., 2021), such as sampling the subset of the multiple objectives. However, it is not suitable in TTA since we need to adapt the model over all instances in the batch.

We introduce a practical technique to minimize the computational load on the solver during TTA. First, we set the number of groups, denoted as $ng$, such that $ng \leq B$ when $B$ is the batch size $B$. Afterwards, we randomly partition the instances within the batch into $ng$ groups, and we utilize the group-averaged losses to enumerate $\nabla_\epsilon \ell(x_i; \theta + \epsilon)$ in Eq. 10 and $\nabla_\theta \ell(x_i; \theta + \hat{\epsilon})$ in Eq. 7. This approach reduces the computational overhead from $O(B)$ to $O(ng)$, and subsequently, the computed weights are evenly distributed among the instances within each group, followed by normalization to ensure they sum up to one. We provide the impact of $ng$ towards the performance from the sensitivity analysis of Figure 4 in Appendix. Algorithm of our proposed model update is in Appendix A.1.

# 4 EXPERIMENTS

## 4.1 EXPERIMENTAL SETTINGS

**Datasets** We evaluate the proposed approaches using three benchmark datasets (Hendrycks & Dietterich, 2019) for test-time adaptation: CIFAR10-to-CIFAR10C, CIFAR100-to-CIFAR100C, and ImageNet-to-ImageNetC, all designed for image classification tasks. For instance, the CIFAR10-to-CIFAR10C scenario simulates adapting the original model, initially trained on CIFAR10 (Krizhevsky et al., 2009), to a stream of test batches from CIFAR10C. In this context, CIFAR10C represents a corrupted version of CIFAR10, which originates from evaluating the robustness of classification networks. Each corrupted dataset contains 15 types of corruptions from four main categories with 5 levels of severity: noise, blur, weather, and digital, all applied to the original dataset. CIFAR100C and ImageNetC have same types of corruptions with CIFAR10C.

**Task** We focus on the online test-time adaptation task, which entails the non-resetting of model parameters after updating them in the current batch, followed by (Wang et al., 2020). We evaluate the proposed model and baselines on CIFAR10-to-CIFAR10C, CIFAR100-to-CIFAR100C, and ImageNet-to-ImageNetC as this standard online setting. Followed by SAR (Niu et al., 2023), we

Table 3: Accuracy (%) on CIFAR10-C (left) and CIFAR100-C (right) under standard online adaptation with WideResNet-28. The improvement by ours are marked as bold in each batch size setting. The underline represents the best results on each case. We reported average accuracy over 15 corruption types.

| CIFAR10-C | Batch size | | | | CIFAR100-C | Batch size | | | |
|---|---|---|---|---|---|---|---|---|---|
| | 8 | 16 | 32 | 64 | | 8 | 16 | 32 | 64 |
| Source | 56.5 | 56.5 | 56.5 | 56.5 | Source | 51.9 | 51.9 | 51.9 | 51.9 |
| TENT | 67.0 | 78.6 | 81.2 | 81.1 | TENT | 5.8 | 19.4 | 53.1 | 66.6 |
| w/ Pareto | **74.8** | **79.6** | 81.1 | **81.3** | w/ Pareto | **27.9** | **44.4** | **60.9** | **67.7** |
| w/ ParetoSAM | 74.1 | **80.0** | **81.3** | **81.4** | w/ ParetoSAM | **21.5** | **40.0** | **60.7** | **67.6** |
| SAR | 64.3 | 78.1 | 81.1 | 81.5 | SAR | 4.0 | 23.7 | 58.9 | 67.4 |
| w/ ParetoSAM | **72.7** | **79.1** | 81.0 | **82.0** | w/ ParetoSAM | **13.8** | **47.8** | **61.2** | **67.5** |

evaluate the models on ImageNet-to-ImageNetC for extreme distribution shift environment, such as the label distribution shift and mixed shift. The label distribution shift means using label distribution changing by timesteps and its label proportions. The mixed shift specifies a model to use a batch sampled from a mixture distribution of 15 corruption types, evaluating robustness on multiple shifted domain. The details are in Appendix A.2

**Baselines** We compare our Pareto-based approach to entropy-based methods as a base structure of TTA. Then, the entropy loss of an instance becomes an individual objective. We first apply the Pareto-orient optimization to TENT (Wang et al., 2020) to show the effects of reducing the conflicts over individual objectives, denoted as TENT w/ Pareto. Additionally, we apply the sharpness-aware Pareto-oriented optimization to TENT for generalization ability, denoted as TENT w/ ParetoSAM. Moreover, we provide modified SAR to incorporate the Pareto-oriented optimization, denoted as SAR w/ ParetoSAM. We provide the details of the implementation in Appendix A.2.

## 4.2 Experimental Results

**Standard Online Adaptation** Table 1 shows the accuracy on ImageNet-C (with severity level 5) under a standard online adaptation setting. When our Pareto-oriented optimization (w/ Pareto, w/ ParetoSAM) improves the baseline performances, we indicated such improvement by bold fonts. TENT shows the lower performance than source pre-trained model, which means that TENT degrades as adaptation proceeds. Compared to the vanilla version of TENT, incorporating Pareto and ParetoSAM into TENT significantly improve the performance of TENT. The proposed approaches update the model by improving all individual objectives without conflicts. Therefore, this Pareto-oriented optimization prevents from degrading the model performance over adaptation procedure. In addition, when we apply the proposed approach to SAR, SAR with ParetoSAM enhances the performances with statistically significant gains. SAR with ParetoSAM finds the perturbation direction to individually maximize all individual objectives while the original version of SAR finds the perturbation direction to maximize the averaged loss regardless of the conflict. Therefore, these results imply that the perturbation direction should take into account individual objectives in order to reduce conflicts and enhance robustness.

**Performance on Small Testing Batches** Table 3 represents the accuracy on CIFAR10-to-CIFAR10C and CIFAR100-to-CIFAR100C (with severity level 5) with various batch sizes. Similar to Table 1, Pareto-oriented approaches improve the baseline performances over batch sizes. TENT is not robust when using small batch, since it tends to collapse to a trivial solution (i.e., predict all instances to the same class) Wang et al. (2020). Therefore, as the batch size decreases, the performance of TENT decreases in both CIFAR10-C and CIFAR100-C. The variations with Pareto and ParetoSAM provides larger performance gains than the original TENT as the batch size becomes smaller. SAR also encounters performance drops in small batch size, as well. The ParetoSAM applied to SAR consistently outperforms compared to the original SAR, and the improvement gain is larger when the batch size is smaller.

**Extreme Distribution Shifts** Table 2 shows the accuracy performance on ImageNet-C (with severity level 5) under online imbalanced label shifts. This setting contains the real-world problem of label distribution changing by timesteps and its label proportions. On-

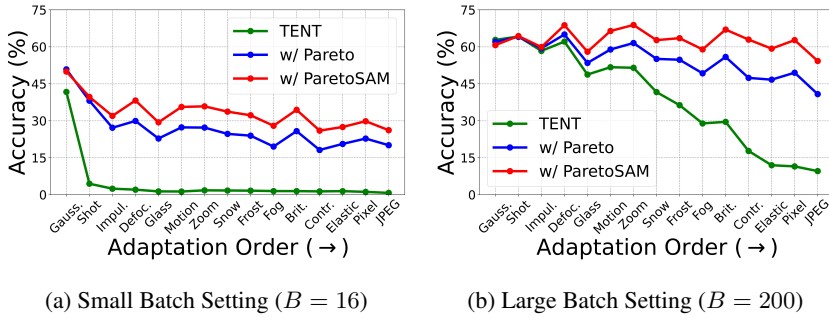

(a) Small Batch Setting ($B = 16$)   (b) Large Batch Setting ($B = 200$)

Figure 2: Accuracy (%) of CIFAR100-C (severity level 5) under continual adaptation setting.

line updating of TTA models can become unstable due to distribution shifts. Therefore, the performance of SAR, SAM-based entropy loss, is better than source and TENT. It indicates the necessity of flatness in real-world TTA problems. SAR with ParetoSAM outperforms SAR with statistically significant improvement. Therefore, these results imply that perturbation for flatness should have a direction that maximizes all individual objectives, not an aggregated objective. We provide the detailed ablation studies in terms of SAM and ParetoSAM in the next subsection. Table 4 shows the accuracy on ImageNet-C (with severity level 5 and 3) under mixture of 15 corruption types. Similar to Table 2, Pareto and ParetoSAM consistently outperform the baselines, which indicates that Pareto-oriented optimization is robust to distribution shift, even in mixed environment.

Table 4: Accuracy (%) on ImageNet-C under mixture of 15 corruption types.

| Method | Level 5 | Level 3 |
|---|---|---|
| Source | 30.6 | 54.0 |
| MEMO | 31.2 | 54.5 |
| DDA | 35.1 | 52.3 |
| TENT | 13.4 | 33.1 |
| **w/ Pareto** | 29.1±0.14 | **54.6**±0.04 |
| **w/ ParetoSAM** | **29.4**±0.05 | **54.6**±0.01 |
| EATA | 38.1 | 56.1 |
| SAR | 38.3 | 57.4 |
| **w/ ParetoSAM** | **39.2**±0.16 | **58.4**±0.04 |

**Reasoning of Robust Performance from Pareto Optimality**
We conjecture that the Pareto-oriented optimization prevents TTA to be degraded into a simple biased classifier (i.e. always predicting a majority class) because the Pareto optimality preserves the individual gradient signal. The preserved individual gradient becomes particularly important when a class becomes minority. Also, a small batch size would turn a class into a minority or a majority class with a small skewness of instance distribution. The same effect can be caused by a biased testing sampler as discussed in experiments on extreme distribution shifts. Given these rational, we observe that the Pareto optimality variations operate robustly under small batch sizes and biased testing sample samplers.

**Analysis on Continual Adaptation**   TTA methods update the model based on the unlabeled loss, which can lead to error accumulation due to noisy predictions as adaptation progresses. To demonstrate the effectiveness of mitigating this error accumulation, we expand our experiments to include continual adaptation, where model parameters are not being reset after each corruption task. In the continual setting, Figure 2 shows the performances of TENT, TENT with Pareto, and TENT with ParetoSAM on CIFAR100-C with small (16) and large (200) batch size. Similar to the previous results in Table 3, TENT shows performance degradation in the early stages, particularly in small batches, due to accelerated collapse. However, in the case of our proposed approach, we observed that less degradation occurs even in continually adapting setting. In large batch case, TENT initially performs well, but eventually shows decline in performances. Conversely, the proposed models, Pareto and ParetoSAM, consistently outperform the baseline and maintain higher performance through the continual adaptation process.

**Conflict Analysis of Perturbation Direction**   Figure 3 shows the visualization of the cosine similarity between the perturbation direction $\hat{\epsilon}$ and individual maximally perturbed direction in the batch. In order to compare the effectiveness of ParetoSAM, we provide the cosine similarity by TENT+ParetoSAM and TENT+SAM. The main difference is that SAM find $\hat{\epsilon}$ from averaged losses, and ParetoSAM is designed to pursue Pareto-Optimality. As shown in Figure 3a, the perturbation by SAM has many conflicts to individual directions. It indicates that the maximal perturbed direction

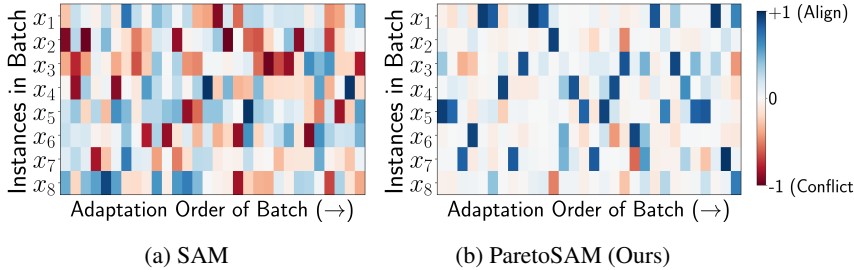

(a) SAM             (b) ParetoSAM (Ours)

Figure 3: Visualization of cosine similarity between the actual perturbation direction $\hat{\epsilon}$ and individually maximal perturbation direction in the batch, where (a) $\hat{\epsilon}$ is from the averaged loss by SAM, and (b) $\hat{\epsilon}$ is toward the proposed Pareto-Optimality, by ParetoSAM. The analysis is on the last batch of Gaussian noise corruption in CIFAR10-C under standard online setting with $\rho = 0.05$.

on the average loss is not aligning with the maximal perturbed direction on the individual losses. Meanwhile, as shown in Figure 3b, ParetoSAM shows fewer conflicts. It means that Pareto-oriented maximal perturbation effectively find the direction to maximize all individual losses. Therefore, we claim that SAM based perturbation may not be the maximization direction for some individual objectives. In order to provide flatness of all individual objectives, the perturbation should have a direction towards Pareto-Optimality.

**Ablation Study on Perturbation Direction**
We provide ablation studies in order to demonstrate the effectiveness of ParetoSAM, qualitatively. ParetoSAM contains two phases of Pareto-Optimality: 1) to find the perturbation direction $\hat{\epsilon}$, 2) to minimize the objective $\mathcal{L}(\theta + \hat{\epsilon})$. In the first phase, as we discussed at the previous analysis, we find the perturbation direction to provide individual flatness. Given the perturbed direction $\hat{\epsilon}$,

Table 5: Ablation Study on ParetoSAM

|  | Perturbation ($\hat{\epsilon}$) | $\mathcal{L}(\theta + \hat{\epsilon}) \downarrow$ | Avg. |
|---|---|---|---|
| TENT w/ SAM | Average | Average | 65.8 |
| TENT w/ ParetoSAM | Pareto | Average | 66.1 |
|  | Average | Pareto | 74.5 |
|  | **Pareto** | **Pareto** | **75.4** |

we minimize the perturbed objectives $\mathcal{L}(\theta + \hat{\epsilon})$ towards Pareto-Optimality in order to reduce conflicts among individual objectives. In order to analyze the effects of each Pareto-Optimality, we conduct ablation studies by replacing the Pareto objective by the averaged individual objectives (Average), on CIFAR10-C of continual adaptation setting. For example, when we apply Average on both phases, the ablated model becomes equivalent to SAM. Table 5 represents the results of ablating the two phases of 1) Pareto optimal perturbation (see Eq. 11), denoted as Pertrubation ($\hat{\epsilon}$) and 2) Pareto optimal gradient update (see Eq. 13), denoted as $\mathcal{L}(\theta + \hat{\epsilon}) \downarrow$. Compared to TENT with SAM, when we apply Pareto to find $\hat{\epsilon}$, the accuracy increases with small margin. This small gain is caused by still existing conflicts when minimizing averaged objectives. Meanwhile, when we apply Pareto optimization to minimize the objectives, ParetoSAMs outperforms SAM, significantly. These results indicates that 1) minimizing the individual objectives towards Pareto-Optimality is important, and 2) minimizing individual flatness towards Pareto-Optimality improves the model performance compared to the average objective.

## 5 CONCLUSION

We propose a new approach to update the model parameters toward Pareto-Optimality across all individual instances in the scenario of test-time adaptation. Pareto-oriented optimization updates the model by the gradient that is agreeable by all instances in the batch, preventing a negative effect from the gradient conflict among test instances. Particularly, this paper suggests an extended version of the Pareto optimization to anticipate unexpected distribution shifts in the test phase by merging the Pareto optimization into sharpness-aware minimization. Therefore, we propose Sharpness-aware Pareto-Oriented optimization to update a model parameter by pursuing individual objectives and flatness. We demonstrate the effectiveness of the proposed approaches through experiments on three benchmark datasets, and our experiments consistently and significantly supports the merit of the proposed optimization.

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

# A APPENDIX

## A.1 ALGORITHM

Algorithm 1 specifies the procedure of our proposed sharpness-aware Pareto-oriented model update.

---

**Algorithm 1** The proposed Sharpness-aware Pareto-oriented model updates at time $t$

---

1: **Input:** the current batch, $\{x_i\}_{i=1}^B$, the current model parameters $\theta_t$, the learning rate $\eta$.
2:   Compute individual gradients with respect to $\epsilon$, $\{\nabla_\epsilon \ell(x_i; \theta_t + \epsilon)\}_{i=1}^B$.
3:   Compute $(\hat{\alpha}_1, ..., \hat{\alpha}_B)$ by Eq. 10 via Frank-Wolfe Algorithm.
4:   Compute $\hat{\epsilon}$ by Eq. 11
5:   Compute $(\hat{w}_1, ..., \hat{w}_B)$ by 7 with $\nabla_\theta \ell(x_i; \theta + \hat{\epsilon})$.
6:   Compute individual gradients on the perturbed loss with respect to $\theta$, $\{\nabla_\theta \ell(x_i; \theta + \hat{\epsilon})\}_{i=1}^B$.
7:   Update $\theta_{t+1} \leftarrow \theta_t - \eta d$   where   $d = \sum_{i=1}^B \hat{w}_i \nabla_\theta \ell(x_i; \theta_t + \hat{\epsilon})$
8: **Output:** updated parameters $\theta_{t+1}$

---

## A.2 ADDITIONAL EXPERIMENTAL SETTINGS

**Implementation Details**   Our base structure of TTA follows the practice of SAR (Niu et al., 2023). Specifically, the model structure is ResNet50-GN (Wu & He, 2018) on ImageNet-C experiments. For CIFAR10-to-CIFAR10C and CIFAR100-CIFAR100C, we follow the settings from Wang et al. (2020; 2022), using WideResNet-28 (Zagoruyko & Komodakis, 2016). We use SGD optimizer with learning rate of $2.5 \times 10^{-4}$ and momentum of 0.9, and we utilize other hyperparameters of baselines as they utilized. Following the previous studies (Wang et al., 2020; Niu et al., 2023), we adopt the affine parameters of the group normalization layers. We set $ng$ as 8 for all cases of batch sizes, and $\rho$ as 0.05.

**Task**   During testing, corrupted batches are provided to the network in an online manner. We follow the evaluation of adaptation performance as standard online test-time adaptation Wang et al. (2020). In this standard online setting, we assess performance for each corruption type individually (Wang et al., 2020; Niu et al., 2022), which means that we reset the model before adapting to a new corruption type. In the continual setting as an analysis, we iteratively adapt the source pre-trained model across a sequence of corruption types without resetting the model, creating more challenging environment (Wang et al., 2022).

## A.3 EXPERIMENTAL ANLAYSIS

**Ablation Study on Group Size**   In order to verify the efficacy of grouping instances to reduce the computational load on solver for Pareto-Optimality, we conduct the sensitivity study on the different size of groups, on CIFAR100 of continual adaptation setting. Figure 4 shows the performance change by different variations of group sizes when the batch size is 200. As a results, there is a trade-off between the performance and computation time depending on the group size. However, when the group size is small, TENT with Pareto shows the competitive performances. As the group size increases, the proposed model improves significantly. Therefore, we efficiently reduce the computational times while maintaining the model performances.

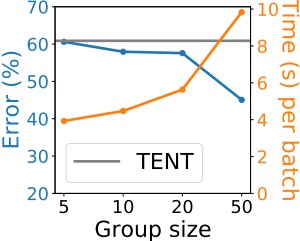

Figure 4: Sensitivity on group size $(ng)$ of TENT w/ Pareto.

