# OpenReview forum: "Towards Pareto-Optimality for Test-Time Adaptation"
_ICLR.cc/2024/Conference — ICLR 2024 Conference Withdrawn Submission_

### Official Review · Reviewer_HqZ3 · 2023-10-27

**Soundness:** 3 good
**Presentation:** 2 fair
**Contribution:** 2 fair
**Rating:** 3
**Confidence:** 5

**Summary:**

This paper proposes to improve test-time adaptation by integrating a so-called pareto-optimality. The underlying idea is that in a minibatch, each instance contributes differently to the gradients, and the average gradients commonly used may be dominated by some instances and make negative contribution. Therefore, this paper proposes to used pareto-optimality to seek an appropriate aggregated gradient instead of average gradients, with instance gradient conflict removed.
Technically, sharpness-aware minimization (SAM) is leveraged and improved by pareto optimization. Experiments show the effectiveness on three benchmark datasets.

**Strengths:**

1. It is interesting to integrate pareto-optimality into the test-time adaptation.
2. Combining pareto-optimality with sharpness-aware minimization is effective.
3. Experiments prove Pareto can work as a plugged-played strategy for improving adaptation performance.

**Weaknesses:**

1. The novelty is limited. Since pareto is a classical multi-objective optimization method and used for multi-task learning, it is not new in this work. Additionall, pareto-optimization has also been used in unsupervised domain adaptation for removing gradient conflict [1], which is even not mentioned in this submission. So, the novelty of the idea is limited.
[1] "pareto domain adaptation" NeurIPS, 2021.
2. The contribution by Integrating Pareto into Sharpness aware minimization is also incremental, which has been discussed in domain generalization.
3. The writting about the specific pareto-optimization algorithm is not clear. This paper focuses more on previous published work, such as SAM, but ignores how to solve the pareto-solutions and alleviate instance gradient conflict in a minibatch. It maynot be enough by only listing the reference.
4. It seems that this paper just applies pareto-optimization in the previous model, and therefore, the novelty in methodology is limited.

**Questions:**

1. It is necessary to mention the closely-related work as citation.
2. The optimization detail is missed.
3. The necessity to combine with SAM is not discussed. The motivation is not strong without clear explannation and intuition behind.

---

### Official Review · Reviewer_7JZo · 2023-10-31

**Soundness:** 2 fair
**Presentation:** 2 fair
**Contribution:** 2 fair
**Rating:** 5
**Confidence:** 4

**Summary:**

This work proposes a novel Pareto-optimality-oriented optimization method to update a pre-trained model for test-time adaptation (TTA). Instead of a simple averaged gradient, it applies the widely-used multiple gradient descent algorithm (MGDA) to find a gradient direction that can minimize the objectives (e.g., entropy losses) of all test instances in a batch. In this way, the model update will not be biased only by a few instances that could lead to worse overall performance. The proposed method is also extended to incorporate the sharpness-aware minimization (SAM) method for better stabilizing the model update.

Experimental results show the proposed method can improve the existing TTA algorithms' performance on three classification benchmarks (CIFAR10-C, CIFAR100-C, and ImageNet-C).

**Strengths:**

+ This work is well-written and easy to follow.

+ Test-time adaption (TTA) is important for real-world applications, and it is interesting to see that TTA can be formulated as a multi-objective optimization problem.

+ The proposed Pareto-optimality-oriented method can achieve promising performance on different TTA classification benchmarks, but also with some concerns (see weaknesses).

**Weaknesses:**

**1. Multi-Objective Optimization with Many Objectives**

The proposed method formulates the TTA as a multi-objective optimization problem with a large number of objectives (B as the batch size), which is also called a many-objective optimization problem. It is well-known that the Pareto set/front (e.g., the set contains all Pareto solutions) is a $(B-1)$-dimensional manifold in non-trivial case, and a vast majority of solutions will be non-dominated (e.g., all be Pareto optimal) with each other when B is large. In other words, it could be hard to find a single gradient direction to improve all B objectives at the same time.

Can the proposed method always find a valid gradient direction via MGDA, especially for a large B and TTA in the wild (e.g., the objectives are quite different and conflicted)?

**2. Computational Overhead**

The proposed method needs to solve two constrained optimization problems (eq 7 and eq 10) with B variables at each step to find a valid gradient direction, which could lead to a long runtime, especially when B is large. A practical grouping method has been developed to reduce the huge computational load. However, the obtained gradient direction is not guaranteed to be valid for all objectives and might lead to worse performance.

- **Grouping and Runtime:** According to the ablation study in Appendix A.3, for CIFAR100-C with batch size 200, the proposed method with group size 5-50 requires a 4s-10s runtime for each batch. Does it mean the total runtime should be 1,000s - 2,500s for handling 50k images? According to [3], the current TTA methods (e.g., Tent [1], EATA [2], and SAR [3]) only need around 100s for processing 50k images for ImageNet-C, which could be even faster for CIFAR100-C. It seems that the proposed method needs at least 10x to 25x computational load than other TTAs. If it is true, with group size 5, the proposed method has similar performance with Tent while requiring 10x run time. Please correct me if I misunderstood something here.

- **Runtime Comparison:** What is the group size of the proposed method used for CIFAR10-C, CIFAR100-C, and ImageNet-C in the main paper? Please report the runtime for each TTA algorithm for all main experiments. If the proposed method requires a significantly larger runtime, it is also interesting to see the performance of Tent/EATA/SAR with a similar computational budget (e.g., multiple runs of gradient updates for each batch).

**3. MGDA for $\epsilon$**

The motivation of MGDA for eq (7) is intuitive, but it is not for eq (10). What is the benefit of finding the same $\epsilon$ via MGDA for all losses in eq (10)? Why not find separate $\epsilon$s for each individual loss via standard SAM?

**4. Novelty and Related Work**

The multi-objective optimization formulation of TTA and SAM proposed in this work is novel based on my understanding. However, the statement "(this work is) the first proposal of merging two concepts of flat-minima and Pareto optimality" is a bit overclaimed. For example, the work [4] also discusses the combination of Pareto optimality and flat-minima from a different perspective (e.g., mode connectivity). In addition, the multi-objective optimization and Pareto optimality have also been investigated for domain adaption [5] and out-of-distribution generalization [6], which could be (not directly) related to this work.

Once the TTA problem is formulated as a multi-objective optimization problem, the proposed method is a straightforward application of MGDA on top of the standard TTA methods (e.g., Tent/EATA/SAR), where the SAM approach is directly from SAR[3].

**5. TTA Problems Other than Classification**

Will the high computational overhead make it very hard for the proposed method to tackle TTA problems other than classification, such as those with dense predictions (e.g., segmentation in Tent[1])?

**Questions:**

- Please address the concerns raised in the above weaknesses.

- In Figure 1, why are some gradients still conflicted with the updating direction found by the proposed method?

- It is better to cite [7,8] for the original MGDA, while Sener & Koltun 2018 is a seminal work on MGDA for multi-task learning.


**Reference**

[1] Tent: Fully Test-Time Adaptation by Entropy Minimization. ICLR 2021

[2] Efficient Test-Time Model Adaptation without Forgetting. ICML 2022.

[3] Towards Stable Test-Time Adaptation in Dynamic Wild World. ICLR 2023.

[4] Pareto Manifold Learning: Tackling multiple tasks via ensembles of single-task models. ICML 2023.

[5] Pareto Domain Adaptation. NeurIPS 2021.

[6] Pareto Invariant Risk Minimization: Towards Mitigating the Optimization Dilemma in Out-of-Distribution Generalization. ICLR 2023.

[7] Steepest descent methods for multicriteria optimization. Mathematical Methods of Operations Research 2000.

[8] Multiple-gradient descent algorithm (MGDA) for multiobjective optimization. Comptes Rendus Mathematique 2012.

---

### Official Review · Reviewer_gRux · 2023-11-01

**Soundness:** 3 good
**Presentation:** 3 good
**Contribution:** 3 good
**Rating:** 6
**Confidence:** 3

**Summary:**

This paper discusses the challenges faced by deep neural networks (DNNs) when dealing with distribution shifts, which occur when there are environmental or noise-related changes between the training and testing data. This can lead to poor performance and unexpected decisions from pretrained models. To address this, it proposes a new gradient-based approach called Test-Time Adaptation (TTA) that uses Pareto-Optimality across the batch to ensure efficient and effective adaption. The paper also introduces a new approach that combines TTA with Sharpness-Aware Minimization (SAM) to better handle extreme scenarios such as mixed distribution shifts, or small batch sizes.

**Strengths:**

- The paper provides a detailed and comprehensive methodology to tackle the issue of distributional shifts in deep neural networks. The Test-Time Adaptation (TTA) method is a novel approach and it provides a solution to a practical problem in Machine Learning.

- The introduction of Pareto-Optimality brings about substantial reduction in the conflicts between individual losses and contributes to effective model adaptations.

- The paper integrates two pivotal concepts, the flat-minima and Pareto optimality, making a significant contribution to the existing literature.

- The robustness of the Pareto-oriented optimization under small batch sizes and biased testing sample samplers is another positive aspect of the research.

- Extensive and thorough experimental validation bolsters the paper's claims and provides ample empirical evidence for the effectiveness of the proposed approaches.

**Weaknesses:**

- The paper relies heavily on established methods such as MGDA and SAM, which could limit its originality and innovation. While these methodologies are well-regarded, building directly upon them may restrict the development of unique approaches.

- Despite the incorporation of Pareto-Optimality and the description of Test-Time Adaptation (TTA) as a continuous adaptation method, the paper lacks thorough examination of the negative effects that emerge when conflicts are disregarded.

- While the research attempts to address practical, real-world adaptations, it fails to provide in-depth understanding of the potential pitfalls and limitations of the proposed methods in different scenarios or with unconsidered factors.

**Questions:**

- Could the authors clarify the practical implications of this research, particularly for industries or sectors beyond academia?

- How does the method respond to multi-modal distribution shifts, and could it be extended to handle these scenarios?

- The research paper has extended the study to 'sharpness-aware Pareto-oriented gradient optimization'. How does this newly proposed approach compliment current SAM based researches?

- The paper does not fully address how negative effects associated with ignoring conflicts during continual adaptation are mitigated. Could you provide more details on this aspect?

---

### Official Review · Reviewer_MZwq · 2023-11-01

**Soundness:** 2 fair
**Presentation:** 3 good
**Contribution:** 2 fair
**Rating:** 3
**Confidence:** 4

**Summary:**

The paper proposes to use multi-gradient descent to address gradient conflicts between samples within each batch during test-time adaptation. This approach is then combined with sharpness-aware minimization to further enhance performance. The effectiveness of the proposed algorithm is demonstrated through experiments on three benchmark datasets.

**Strengths:**

1. Considering gradient conflicts during test-time adaptation is interesting.

2. The algorithm description is detailed and clear, and the proposed method shows promising performance on three datasets.

**Weaknesses:**

1. The application of MGDA in test-time adaptation is straightforward, and the novelty of the approach is limited.

2. The authors aim to optimize towards Pareto-Optimality, but it is unlikely that a single step of MGDA can achieve this. Convergence towards Pareto-Optimality in the original MGDA requires many steps of gradient descent. The paper should provide theoretical proof or justification as to why the proposed direction is closer to Pareto optimal than the average direction. Additionally, the proposed method does not address conflicts between different batches.

3. Some papers (e.g., [1, 2]) have shown that the performance improvement of MGDA is due to regularization effects rather than resolving gradient conflicts. Weighted (or unweighted) average with regularization has demonstrated similar or better performance. The authors should clearly show that the performance improvement in their method is a result of solving gradient conflicts (towards Pareto Optimal) rather than regularization.

[1] In Defense of the Unitary Scalarization for Deep Multi-Task Learning, NeurIPS 2022.

[2] Scalarization for Multi-Task and Multi-Domain Learning at Scale, NeurIPS 2023.

**Questions:**

See Weaknesses.